# Colistin Resistance and ESBL Production in *Salmonella* and *Escherichia coli* from Pigs and Pork in the Thailand, Cambodia, Lao PDR, and Myanmar Border Area

**DOI:** 10.3390/antibiotics10060657

**Published:** 2021-05-31

**Authors:** Khin Khin Lay, Saharuetai Jeamsripong, Kyaw Phyoe Sunn, Sunpetch Angkititrakul, Ransiya Prathan, Songsak Srisanga, Rungtip Chuanchuen

**Affiliations:** 1Department of Animal Science, University of Veterinary Science, Nay Phi Tau 15013, Myanmar; khinkhinvph@gmail.com; 2Research Unit in Microbial Food Safety and Antimicrobial Resistance, Department of Veterinary Public Health, Faculty of Veterinary Science, Chulalongkorn University, Bangkok 10330, Thailand; saharuetai.j@chula.ac.th (S.J.); drkyawphyosunn@gmail.com (K.P.S.); rangsiya.p@chula.ac.th (R.P.); songsak.s@chula.ac.th (S.S.); 3Division of Public Health, Department of Livestock Breeding and Veterinary, Ministry of Agriculture, Livestock and Irrigation, Nay Phi Tau 15011, Myanmar; 4Research Group for Prevention Technology in Livestock, Faculty of Veterinary Medicine, Khon Kaen University, Khon Kaen 40002, Thailand; sunpetch@kku.ac.th

**Keywords:** *Escherichia coli*, colistin resistance, extended-spectrum β-lactamase, *Salmonella*, southeast Asia

## Abstract

The study aimed to examine the prevalence and genetic characteristics of ESBL-production and colistin resistance in *Salmonella* and *Escherichia coli* from pigs and pork in the border area among Thailand, Cambodia, Lao PDR, and Myanmar. *Salmonella* (*n* = 463) and *E. coli* (*n* = 767) isolates were collected from pig rectal swab from slaughterhouses (*n* = 441) and pork from retail markets (*n* = 368) during October 2017 and March 2018. All were determined for susceptibility to colistin and cephalosporins, ESBL production and *mcr* and ESBL genes. *Salmonella* was predominantly found in Cambodia (65.8%). Serovars Rissen (35.6%) and Anatum (15.3%) were the most common. The *E. coli* prevalence in pork was above 91% in all countries. Colistin-resistance rate in *E. coli* (10.4%) was significantly higher than *Salmonella* (2.6%). ESBL-producing *Salmonella* (1.9%) and *E. coli* (6.3%) were detected. The *bla*_CTX-M-55_ and *bla*_CTX-M-14_ were identified. The *mcr-1* gene was detected in *Salmonella* (*n* = 12) and *E. coli* (*n* = 68). The *mcr-1*/*bla*_CTX-M-55_ and *mcr-3*/*bla*_CTX-M-55_ co-concurrence was observed in one *Salmonella* and three *E. coli* isolates, respectively. In conclusion, pigs and pork serve as carriers of colistin and new generation cephalosporins resistance. Testing for resistance to last line antibiotics should be included in national AMR surveillance program using One Health approach.

## 1. Introduction

Multidrug resistance (MDR) in bacterial pathogens has escalated and become a significant cause of morbidity and mortality worldwide [1]. Resistance to last resort antibiotics (e.g., colistin and newer generation cephalosporins) is of particular concern because it may limit antibiotic therapy for bacterial infections in the near future [2,3].

Cephalosporins (3rd, 4th, and 5th generations) are used for disease treatment in humans and animals [4] and categorized as critically important antimicrobials in both human [5] and veterinary medicine [6]. Over recent years, resistance to cephalosporins have been increasingly reported in humans and animals [7,8] and role of livestock as important sources of extended-spectrum β-lactamase (ESBL) producers potentially spread to humans through the food chain was highlighted [9]. Resistance to cephalosporins is mainly due to the production of ESBLs that confer resistance to most β-lactams, including new generation cephalosporins and monobactams (e.g., aztreonam), but not cephamycin (e.g., cefoxitin) and carbapenems (e.g., imipenem) [10]. ESBL-encoding genes are primarily located on plasmid that can be horizontally transferred to inter and intra bacterial species. The genes frequently co-exist with other resistance genes on the same large plasmids, resulting in MDR phenotype of ESBL producers [9,11]. Colistin or polymyxin E has a narrow antibacterial spectrum mainly against Gram-negative bacteria [12]. Colistin is widely used in pig production for treatment and control of *Enterobacteriaceae* infections. Colistin-resistant *E. coli* carrying plasmid-borne *mcr-1* were isolated from food animals in China in 2016 [13]. Since then, several *mcr* variants have been identified in livestock in many parts of the world, including countries with zero to marginal use of colistin (e.g., the United States) [14]. The *mcr* variants can occur due to transference of multiple mobile genetic elements. The co-occurrence of variants of *mcr* genes and other antimicrobial resistance (AMR) genes can increase the widespread of colistin resistance and pose a global public health risk [15].The border-crossing areas between Thailand and its land neighboring countries including Cambodia, Lao PDR and Myanmar has flourished and become a major route for cross-border trade and tourism owing to their geographical advantages situated along the Mekong river [16]. Purchasing and selling live animals and retail meat, especially pigs and pig products, is one of the high-trade volume and value. The cross-border trade has been emphasized on improvement of their commerce, investment, and economic cooperation [16]. This raises a particular concern over the role of food animals and products as reservoirs of diseases and AMR bacteria that possibly spread to other regions and countries. 

Currently, knowledge on resistance to new generation cephalosporins and colistin in bacteria from livestock and products remains largely undiscovered in most cross-border regions globally. This study aimed to examine the prevalence and genetic characteristics of ESBL-production and colistin resistance in *Salmonella* and *E. coli* isolated from pigs and pork in the border area between Thailand, Cambodia, Lao PDR, and Myanmar.

## 2. Results

### 2.1. Salmonella Prevalence and Serovars in Pigs and Pork

Half of the samples were positive to *Salmonella* (49.8%) (Table 1). The highest *Salmonella* prevalence was found in Cambodia (65.8%, 98/149), followed by Thailand (56.3%, 214/380). The *Salmonella* positive samples in retail markets (61.0%, 246/403) was 1.6 times greater than those in slaughterhouses (39.0%, 157/403). A total of 463 *Salmonella* isolates with 60 serovars were obtained from Thailand (*n* = 237), Cambodia (*n* = 121), Lao PDR (*n* = 86) and Myanmar (*n* = 19) (Table 2). The most common *Salmonella* serovars were Rissen (35.6%), followed by Anatum (15.3%). 

### 2.2. E. coli Prevalence in Pork

The overall percentage of *E. coli* in pork from the markets was 44.7% (343/767). The high prevalence (>88%) was observed in pork in all countries. The high recovery rate (>94%) for commensal *E. coli* was observed for rectal swab sample, indicating the high competence of *E. coli* isolation method used (Table 1). Subtotal of *E. coli* was not identified in Table 1 to avoid the comparison among the sampling sources since commensal *E. coli* regularly present in rectal swab. Seven hundred and sixty-seven isolates were collected from Thailand (*n* = 368), Cambodia (*n* = 136), Laos (*n* = 133), and Myanmar (*n* = 130) for further experiments (Table 1).

### 2.3. ESBL Production and Colistin Resistance in Salmonella and E. coli

Of all *Salmonella*, colistin-resistant *Salmonella* was found at low frequency (2.8%) (Table 3). Colistin-resistant *Salmonella* were found in pork from Myanmar (5.6%), Cambodia (5.3%), Thailand (4.2%), and Laos (1.8%). Cephalsosporin-resistance rates were still low (<2.4%), of which the Thai pork isolates exhibited the highest prevalence to all cephalosporins tested (Table 4). 

For *E. coli*, overall colistin-resistance rate (10.4%) was significantly higher (*p* < 0.05) than that in *Salmonella* (Table 3 and Table 4). Colistin resistance was most common in Lao PDR (28.6%, 38/133), where the highest rate was observed among the pork isolates (37.0%, 20/54). Resistance rates to ceftazidime (6.4%) cefotaxime (9.0%) and cefpodoxime (8.7%) in *E. coli* were higher than those in *Salmonella* (Table 4).

### 2.4. Prevalence of ESBL-Producing Salmonella and E. coli

The ESBL-producing *Salmonella* percentage was low (1.9%, 9/463). The isolates were found only in Thailand (3.4%, 8/237) and Lao PDR (1.2%, 2/86) (Table 3).

The overall prevalence of ESBL-producing *E. coli* was 6.5% (50/767), of which the highest percentage was among the Myanmar isolates (13.9%, 18/130) (Table 3).

The prevalence of ESBL-producing *E. coli* (6.5%) was significantly different from that of *Salmonella* (1.9%). The presence of ESBL-producing *E. coli* was associated with sample type, and significantly higher in pigs (7.8%, 33/424) than pork (5.0%, 17/343) (OR = 2.88, *p* = 0.028). A *Salmonella* isolate (from Laos) and 4 *E. coli* from Thailand (*n* = 2) and Cambodia (*n* = 2) were ESBL producers that were additonally resistant to colistin.

### 2.5. Genotype of Colistin-Resistant and ESBL-Producing Salmonella and E. coli

In *Salmonella*, all colistin-resistant isolates (*n* = 13) carried *mcr-1* (Table 3). A Thai pig isolate additionally harbored *mcr-3*. Only *bla*_CTX-M55_ (88.9%, 8/9) was observed in ESBL-producing *Salmonella* (Table 3). The only ESBL-producing *Salmonella* from pork in Lao PDR carried both *mcr-1* and *bla*_CTX-M55_.

In *E. coli*, *mcr-1* (85%, 68/80) and *mcr-3* (38.8%, 31/80) were found (Table 3). Five *mcr-1*-possitive *E. coli* from three Thai and two Cambodian pig isolates were susceptible to colistin. Six colistin-resistant *E. coli* from pigs in Thailand (*n* = 1) and Cambodia (*n* = 1) and pork in Camodia (*n* = 1) and Lao PDR (*n* = 3) carried both *mcr-1* and *mcr-3* (colistin MIC = 4 and 8 µg/mL). The *bla*_CTX-M-55_ (70.8%, 34/48) and *bla*_CTX-M-9_ (32%, 16/48) genes were detected among the ESBL-producing *E. coli*. Three *E. coli* from Thai pig (*n* = 1) and Cambodian pig (*n* = 1) and pork (*n* = 1) harbored both *mcr-3* and *bla*_CTX-M55_. 

The β-lactamase gene, *bla*_TEM-1_, was common in ESBL-producing *Salmonella* (77.8%, 7/9) and *E. coli* (54.2%, 26/48). The percentage of ESBL production was statistically associated with *bla*_TEM-1_ (*p* < 0.05).

The *bla*_CTX-M55_ gene in one ESBL-producing *Salmonella* from Thai pig and seven ESBL-producing *E. coli* from Cambodia (one from pig and one from pork) and Myanmar (three from pigs and two from pork) was conjugally transferred.

## 3. Discussion

A major finding in this study was the high prevalence of *E. coli* contamination in pork in retail markets, in agreement with a previous study [17]. Concurrently, the *Salmonella* contamination rate was approximately 1.9 times lower than that of *E. coli*. These observations emphasize that legal framework of microbiological quality of raw meat must be effectively implemented for food safety in retail market facilities among border provinces. In general, retail meat is delivered in open buckets and sold in open-air under ambient temperature. Unsold meat is stored overnight in an icebox for re-selling in the following day. These unhygienic practices contribute to the high bacterial contamination observed. Importantly, *Salmonella* contamination increased at retail markets, suggesting cross-contamination during slaughtering process or transportation or from environment at retail markets. However, the samples were not collected immediately after slaughtering and before transporting to retail markets, therefore, the hypothesis cannot be scrutinized. Together with the high *E. coli* contamination, it indicates that fresh meat at retail markets serve as a major source of foodborne pathogens and highlights the need for the effective-hygienic practice along food production chain. 

Thailand has launched law and regulations to contain *Salmonella* in poultry since 2010 [18]. The law includes the enforcement of sample collection for *Salmonella* detection in broiler, laying hens and breeders. Such law has not been specifically issued for pig farm. However, the microbiological quality of meat is regulated under the Microbiological Quality Criteria for Food and Food Contact Containers No.3 (2017), indicating that *Salmonella* must not be present in 25 g of fresh and frozen raw meat (either pork or chicken meat) intended for human consumption [19]. Therefore, good manufacturing practice is required at both farm and slaughtering process levels to produce *Salmonella* free raw meat.

In this study, the most common *Salmonella* serovars were Rissen and Anatum, in agreement with a previous study [20]. *Salmonella* Enteritidis was not observed in this study. This maybe because the major reservoir of *Salmonella* Enteritidis is not in swine production in agreement of previous study [17]. ESBL-producing *Salmonella* and *E. coli* were present at lower rate when compared to previous studies in Thailand, [21,22] and other countries e.g., Denmark [23], Czech [24], Nigeria [25], and Germany [26]. The differences could be associated with many factors (e.g., sampling period, geographical variations and different antimicrobials usage). This study covered the border provinces of Thailand and 3 neighboring countries (i.e., Cambodia, Lao PDR and Myanmar). However, when compared to our previous study conducted in Thailand-Cambodia border province, the prevalence of ESBL-positive *Salmonella* (2.2%) in this study was higher than that observed (1.7%) in the same area [27]. Due to high cost of cephalosporins, use of these antimicrobials in livestock production will increase investment cost and reduce profits. Therefore, cephalosporins may not be commonly used in food-animals in the study areas. In addition, distribution of ESBL producers is not always a direct effect of cephalosporin usage [28] and could be due to coselection of ESBL genes by other antibiotics, e.g., ampicillins, aminoglycosides, tetracyclines that are commonly used in swine production in Thailand [29].

Most ESBL-producing *Salmonella* (8%) and *E. coli* (54%) carried *bla*_TEM-1_ and almost all carried CTX-M group 1 and 9, in agreement with a previous study showing that *bla*_CTX-M-positive_
*E. coli* isolates usually non ESBL *bla*_TEM-1_ [30]. This was also supported by the statistical significance of association between ESBL production and *bla*_TEM-1_ (*p* < 0.05) observed. In addition, not all ESBL producers in this study carried the ESBL genes tested, indicating the existence of other resistance mechanisms and/or ESBL genes that were not included in this study.

A previous study in the same region demonstrated that CTX-M β-lactamases were common among ESBL-producing isolates in this study [31]. This was in agreement with the current study where *bla*_CTX-M-55_ of CTX-M-1 and *bla*_CTX-M-14_ of CTX-M-9 were predominant. Several studies previously reported the common presence of *bla*_CTX-M-1_ in food animals in Switzerland, Tunisia, the United Kingdom, and Germany [9,10,32,33]. The *bla*_CTX-M55_ genes were located on conjugative plasmids, in agreement with our previous studies [27] and supporting its distribution in this area. Horizontal transfer of *bla*_CTX-M-55_ was under selective pressure of ampicillin in conjugation experiment, highlighting the role of the old generation antibiotic that has been widely used for a long time as a selective pressure for the new generation of clinically important antibiotics. In addition, the *bla*_CTX-M-14_ gene was most frequently found in ESBL-producing *E. coli*, in agreement with food animals in China [34] and Denmark [23].

In Thailand, the use of medicated feed containing polymyxins, cephalosporins, fluoroquinolones and others is regulated by the law on “Characteristics and conditions of animal feed containing drugs prohibited from producing, importing, selling and using” that was issued in 2018 [35]. The regulation of antimicrobials drugs (including polymyxins B, colistin and other drugs in penicillin and fluroquinolone groups) that must not be mixed in animal feed for prophylactic purposes was later announced [36]. The use of colistin for short-term treatment can be performed in swine production [37]. Colistin-resistant *E. coli* was found in pigs and pork, reflecting the use of colistin in pig production in the region. Interestingly, colistin-resistant *Salmonella* was limited to pork. This is rather a result of cross contamination during slaughtering, transportation, and in retail markets. 

Most colistin-resistant *Salmonella* and *E. coli* harbored either *mcr-1* or *mcr-3*, supporting the significance of *mcr* genes on spread of colistin resistance worldwide. However, some colistin-resistant isolates lacked both genes, indicating the existence of other colistin-resistance mechanisms not characterized in this study. Colistin-resistant *Salmonella* and *E. coli* harboring both *mcr-1* and *mcr-3* were isolated, in agreement with previous studies [38]. Their colistin MICs were 4 and 8 µg/mL (data not shown) and not different from those with *mcr-1* and *mcr-3* alone, suggesting the variable effect of *mcr* genes on colistin-resistance level. The common presence of *mcr-3* (either alone or with *mcr-1*) has warned about the possibility of its global spread as observed for *mcr-1*. None of *mcr-2* was observed, in agreement with a previous study [39]. 

The findings of *Salmonella* with *mcr-1*/*bla*_CTX-M-55_ (*n* = 1) and *E. coli* with *mcr-3*/*bla*_CTX-M55_ (*n* = 3) are of particular concern. This is consistent with previous studies [40,41]. Despite the low percentage, the co-occurrence generates impact on human and veterinary medicine due to the potential contribution to the spread of bacteria with clinically important antimicrobial resistance, resulting in limited choice for bacterial infection therapy. Resistance genes in these groups are plasmid borne and potentially distribute to bacteria in humans, other animals and environment, so implementation of One Health approach is needed to alleviate the AMR issue.

## 4. Materials and Methods

### 4.1. Sampling Location and Sample Collection

The sampling area are boundary provinces along the border between Thailand and Cambodia (Sa Kaeo and Banteay Meanchey), Thailand and Lao PDR (Nong Khai and Vientiane), and Thailand and Myanmar (Chiang Rai and Tarchileik) (Figure 1). The samples were collected during 2017–2018. All pig samples were obtained from one municipal pig slaughterhouse and one municipal fresh market in each province.

All pigs were from commercial production farms that provided meat for domestic consumption. In Thailand, the pig slaughterhouse in Nong Khai was a large-scale facility with greater than 80 pigs were slaughtered each day, while slaughterhouses in Sa Kaeo and Chiang Rai provinces were small-scale modern facilities with no more than 50 pigs were processed daily. The pig slaughterhouses in Banteay Meanchey province, Cambodia and from Tachileik, Myanmar were traditionally small slaughterhouses with a throughput of 30 or fewer pigs per day. In Lao PDR, the pig slaughterhouse in Vientiane was large-scale modern facility with a throughput of greater than 200 pigs per day. A total of 809 samples were collected by rectal swab from pigs in slaughterhouses (*n* = 441) and carcass swab in retail markets (*n* = 368) from Thailand (*n* = 380), Cambodia (*n* = 149), Lao PDR (*n* = 140) and Myanmar (*n* = 140) (Table 1).

At slaughterhouses, rectal swab samples were obtained by rectal evacuation after bleeding but prior to scalding. At fresh markets, the pig carcass swab was obtained by swabbing an area of at least 50 cm^2^ on each fresh carcass. The sample collection was performed three times at each sampling site. All samples were stored in an icebox and transported within 24 h after collection for immediate processing or held in refrigerator at 4 °C until exanimation 48 h of collection.

### 4.2. Isolation and Identification of Salmonella and E. coli

*Salmonella* was isolated following ISO 6579:2002(E) [42] and subjected to serotyping by slide agglutination based on the Kaufman-White scheme [43]. A single colony of each serotype was collected from each positive sample. *E. coli* was isolated and biochemically confirmed [44,45]. One isolate was collected from each positive sample and all were stored as 20% glycerol stock at −80 °C. 

### 4.3. Determination of Minimum Inhibitory Concentration (MICs) of Colistin

All the *Salmonella* (*n* = 463) and *E. coli* (*n* = 767) isolates were tested for susceptibility to colistin by MIC determination using two-fold agar dilution method [46]. The colistin concentrations ranged from 0.0625 to 64 µg/mL. The clinical breakpoint for defining colistin resistance was 2 µg/mL [47]. Multidrug resistance (MDR) was defined as being resistant to at least three different classes of antibiotics.

### 4.4. Determination of ESBL Production

ESBL production was initially screened and confirmed using disk diffusion method in all the *Salmonella* and *E. coli* isolates [46]. Initial screening was performed using ceftazidime (30 ug), cefotaxime (30 ug) and cefpodoxime (10 ug) (Oxoid, Hampshire, England). All the isolates resistant to at least one of the indicator cephalosporins were subjected to confirm ESBLs production by using combination disk diffusion method with ceftazidime (30 µg) and cefotaxime (30 µg) alone and in combination with clavulanic acid (10 µg) (Oxoid). A difference of ≥5 mm between the inhibition zone of the cephalosporin/clavulanic acid combination and corresponding cephalosporin disks alone was interpreted as positive ESBL phenotype *E. coli* ATCC 25922 was used as a quality control strain. 

### 4.5. PCR and Nucleotide Sequencing

PCR-template DNA was prepared by whole cell boiled lysate procedure [48]. All the primers used are listed in Table 5. All PCR reactions were performed using GeNei™mastermix (Merck, Munich, Germany). PCR amplicons were gel purified using Nucleospin^®^ Gel (Macherey-Nagel, Düren, Germany) and submitted for sequencing using PCR primers at First Base Laboratories (Selangor Darul Ehsan, Malaysia). The obtained DNA sequence was BLAST compared with GenBank database (http://www.ncbi.nlm.nih.gov/ BLAST accessed on 24 May 2021).

### 4.6. Detection of ESBL and mcr Genes

All Salmonella (*n* = 12) and *E. coli* (*n* = 80) resistant to colistin were screened for the presence of *mcr-1* [13], *mcr-2* [39], and *mcr-3* [49]. All the isolates that were confirmed to be ESBL produccers including ESBL-producing *Salmonella* (*n* = 9) and *E. coli* (*n* = 50) were examined for β-lactamase genes including *bla*_TEM_, *bla*_SHV_, *bla*_CMY-1_ [51], and *bla*_CTX-M_ and *bla*_PSE-M_ [50]. CTX-M subgroups including *bla*_CTX-M1_, *bla*_CTX-M2_, *bla*_CTX-M8/25_ [52], and *bla*_CTX-M9_ [53], were further examined in *bla*_CTX-M-positive_
*E. coli*. The *E. coli* isolates positive to *bla*_CTX-M1_ were examined for *bla*_CTX-M15_ [54]. All PCR amplicons were gel purified and submitted for sequencing using PCR primers. The presence of *bla*_CMY-2_ was examined [51].

### 4.7. Test of Conjugation Transfer of ESBL Genes 

Conjugation experiment was conducted in all ESBL producers using biparental mating method [55]. *E. coli* MG1655 (rifr MG1655; MIC = 256 µg/mL) were used as recipient for ESBL-producing *Salmonella* donors and S. Enteritidis SE12 (rifr SE12; MIC = 256 µg/mL) was used as recipient for ESBL-producing *E. coli* donors. All transconjugants selected on LB agar containing rifampicin (32 µg/mL) and ampicillin (100 µg/mL) and were tested for ESBL genes as in correspondent donors. 

### 4.8. Statistical Analysis

The significance differences between the prevalence of *Salmonella* and *E. coli* and the occurrence of AMR stratified by population, location, and sample type were determined using Pearson’s Chi-square or Fisher’s exact test. The statistical analyses were performed with IBM SPSS Statistics for Windows, Version 18.0 (SPSS Inc., Chicago, IL, USA). Logistic regression analysis was used to measure the strength of association between phenotype and genotype of resistance (STATA SE12, Stata Corp LLC.: College Station, TX, USA). A *p*-value of <0.05 was considered statistically significant. Odds ratio and 95% confidence interval (CI) were calculated.

## 5. Conclusions

Healthy pigs can serve as reservoirs for colistin-resistant and ESBL-producing *Salmonella* and *E. coli* that may constitute a public health threat. It highlights the need for surveillance program as well as control and prevention strategic plan for AMR in bacteria of food animal origin at national and regional levels. Standard hygiene and sanitation practices should be enhanced in retail markets to minimize the cross contamination. The epidemiological data will help to understand the root cause of AMR. Guidance of the interventions and evaluation of the success of the inventions from farm to fork is essential for the region.

## Figures and Tables

**Figure 1 antibiotics-10-00657-f001:**
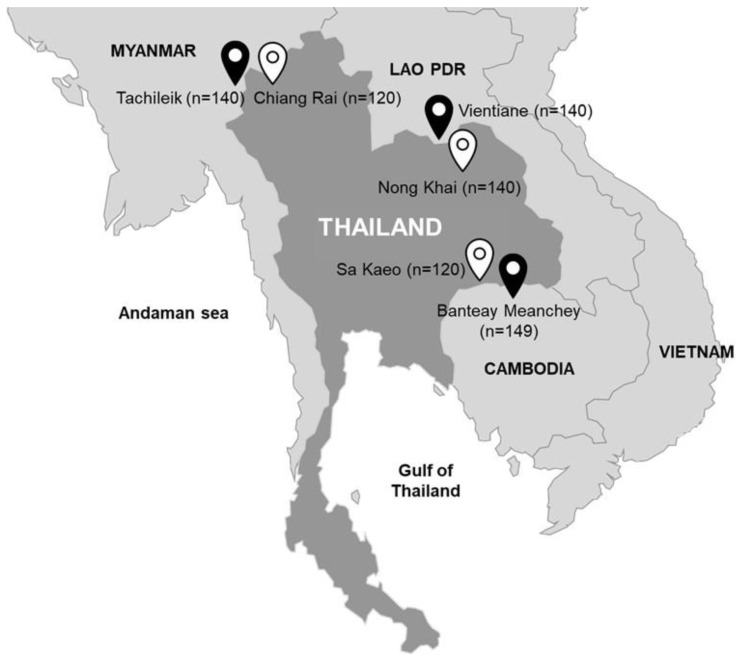
Sampling location in the border provinces between Thailand (
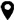
) and other neighboring counties (
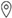
) including Lao PDR, Cambodia and Myanmar. Number of samples is indicated in parenthesis.

**Table 1 antibiotics-10-00657-t001:** Number of samples and prevalence of *Salmonella* and *E. coli* in the border provinces among Thailand, Cambodia, Lao PDR, and Myanmar (*n* = 809).

Country	Location	Source	Type of Sample	No. of Sample	No. of Positive Sample (%)	
					*Salmonella*	*E. coli* ^1^
Thailand	Nong Khai	Slaughterhouse	Rectal swab	80	57 (71.3)	79 (98.8)
		Market	Carcass swab	60	46 (76.7)	57 (95.0)
			Subtotal	140	103 (73.6)	-
	Sa Kaeo	Slaughterhouse	Rectal swab	60	2 (3.3)	58 (96.7)
		Market	Carcass swab	60	34 (56.7)	58 (96.7)
			Subtotal	120	36 (30.0)	-
	Chiangrai	Slaughterhouse	Rectal swab	60	30 (50.0)	58 (96.7)
		Market	Carcass swab	60	45 (75.0)	58 (96.7)
			Subtotal	120	75 (62.5)	-
Cambodia	Banteay Meanchey	Slaughterhouse	Rectal swab	84	39 (46.4)	79 (94.0)
		Market	Carcass swab	65	59 (90.8)	57 (87.7)
			Subtotal	149	98 (65.8)	-
Lao PDR	Vientiane	Slaughterhouse	Rectal swab	82	28 (34.1)	79 (96.3)
		Market	Carcass swab	58	44 (75.9)	54 (93.1)
			Subtotal	140	72 (51.4)	-
Myanmar	Tarchileik	Slaughterhouse	Rectal swab	75	1 (1.3)	71 (94.7)
		Market	Carcass swab	65	18 (27.7)	59 (90.8)
			Subtotal	140	19 (13.6)	-
			Grand total	809	403 (49.8)	343(93) ^2^

^1^ subtotal and total percentage was not calculated as commensal *E. coli* are normally identified in rectal swab. ^2^ calculated for carcass swab in markets only.

**Table 2 antibiotics-10-00657-t002:** *Salmonella* serovars from pig and pork samples (*n* = 468) in Thailand, Lao PDR, Cambodia, and Myanmar.

*Salmonella* Serotype	No. of Isolates (%)							
	Thailand (*n* = 237)		Laos PDR (*n* = 86)		Cambodia (*n* = 121)		Myanmar (*n* = 19)	
	Pig	Pork	Pig	Pork	Pig	Pork	Pig	Pork
Typhimurium	12 (5.0)	-	1 (1.2)	-	2 (1.7)	-	-	-
Sao	12 (5.0)	8 (3.3)	-	-	-	-	-	-
Augustenborg	1 (0.4)	1 (0.4)	-	2 (2.3)	-	-	-	-
Schwarzengrund	2 (0.8)	-	-	1 (1.2)	-	-	-	-
Derby	2 (0.8)	-	-	-	3 (2.5)	1 (0.8)	-	-
Rissen	37 (15.5)	66 (27.6)	-	11 (12.8)	18 (14.9)	30 (24.8)	1 (5.3)	2 (10.5)
Saintpaul	2 (0.8)	8 (3.3)	-	1 (1.2)	-	-	-	2 (10.5)
Eastbourne	1 (0.4)	-	-	-	-	-	-	-
Anatum	8 (3.3)	30 (12.6)	5 (5.8)	5 (5.8)	-	9 (7.4)	-	14 (73.7)
Rideau	1 (0.4)	4 (1.7)	-	1 (1.2)	-	-	-	-
Sanktmarx	5 (2.1)	10 (4.2)	1 (1.2)	2 (2.3)	-	-	-	-
Weltevreden	4 (1.7)	1 (0.4)	-	-	1 (0.8)	1 (0.8)	-	-
Braenderup	1 (0.4)	-	-	-	-	2 (1.7)	-	-
Fareham	4 (1.7)	2 (0.8)	-	-	-	-	-	-
Stanley	2 (0.8)	1 (0.4)	8 (9.3)	7 (8.1)	3 (2.5)	5 (4.1)	-	-
Vijle-1	-	1 (0.4)	-	-	-	-	-	-
Norwich	-	2 (0.8)	-	1 (1.2)	-	-	-	-
Yalding	-	4 (1.7)	-	-	-	-	-	-
Calabar	-	1 (0.4)	-	-	-	-	-	-
Hayindogo	-	3 (1.3)	-	6 (7.0)	-	1 (0.8)	-	-
Muenster	-	1 (0.4)	-	3 (3.5)	1 (0.8)	5 (4.1)	-	-
Potto	-	-	3 (3.5)	-	-	-	-	-
Tsevie	-	-	1 (1.2)	-	2 (1.7)	-	-	-
Brunei	-	-	4 (4.7)	-	-	-	-	-
Kissi	-	-	1 (1.2)	-	-	-	-	-
Eschberg	-	-	1 (1.2)	-	-	-	-	-
Ayinde	-	-	1 (1.2)	-	-	-	-	-
Kentucky	-	-	1 (1.2)	2 (2.3)	-	-	-	-
Rottnest	-	-	1 (1.2)	-	-	-	-	-
Vilvoorde	-	-	1 (1.2)	-	-	-	-	-
Kouka	-	-	1 (1.2)	-	-	-	-	-
Portanigra	-	-	-	1 (1.2)	-	-	-	-
Newlands	-	-	-	2 (2.3)	-	-	-	-
Bristol	-	-	-	1 (1.2)	-	-	-	-
Sandow	-	-	-	1 (1.2)	-	10 (8.3)	-	-
Haifa	-	-	-	1 (1.2)	-	1 (0.8)	-	-
Magumeri	-	-	-	1 (1.2)	-	-	-	-
Lika	-	-	-	1 (1.2)	-	-	-	-
V	-	-	-	1 (1.2)	-	-	-	-
Koenigstuhl	-	-	-	1 (1.2)	1 (0.8)	-	-	-
II	-	-	-	1 (1.2)	-	1 (0.8)	-	-
Suberu	-	-	-	1 (1.2)	-	-	-	-
Ikayi	-	-	-	2 (2.3)	-	-	-	-
Dallgow	-	-	-	-	1 (0.8)	-	-	-
Paratyphi-B	-	-	-	-	1 (0.8)	-	-	-
Lekke	-	-	-	-	1 (0.8)	-	-	-
Herston	-	-	-	-	1 (0.8)	-	-	-
Hvittingfoss/II	-	-	-	-	1 (0.8)	-	-	-
Stanley ville	-	-	-	-	1 (0.8)	-	-	-
Bradford	-	-	-	-	1 (0.8)	-	-	-
Yoruba	-	-	-	-	1 (0.8)	-	-	-
Rechovot	-	-	-	-	4 (3.3)	-	-	-
Bracknell	-	-	-	-	1 (0.8)	-	-	-
Idikan	-	-	-	-	1 (0.8)	-	-	-
Sinstorf	-	-	-	-	-	5 (4.1)	-	-
Paris	-	-	-	-	-	1 (0.8)	-	-
Newport	-	-	-	-	-	1 (0.8)	-	-
Ituri	-	-	-	-	-	1 (0.8)	-	-
Kedougou	-	-	-	-	-	1 (0.8)	-	-
Havana	-	-	-	-	-	1 (0.8)	-	-
Total	94 (39.7)	143 (60.3)	30 (34.9)	56 (65.1)	45 (37.2)	76 (62.8)	1 (5.3)	18 (94.7)

**Table 3 antibiotics-10-00657-t003:** Distribution of phenotypic and genotypic of ESBL production and colistin resistance in *Salmonella* (*n* = 463) and *E. coli* (*n* = 767).

Country	Source	Total No.	*Salmonella*						Source	Total No.	*E. coli*						
			ESBL Production, No. (%) ^1^			Colistin Resistance, No. (%) ^2^					ESBL Production, No. (%) ^1^				Colistin Resistance, No. (%) ^2^		
			Total	*bla* _CTX-M-55_	*bla* _TEM-1_	Total	*mcr-1*	*mcr-3*			Total	*bla* _CTX-M-55_	*bla* _CTX-M-14_	*bla* _TEM-1_	Total	*mcr-1*	*mcr-3*
Thailand	Pig	94	2 (2.1)	1 (1.1)	1 (1.1)	-	-	1 (1.1)	Pig	195	12 (6.2)	8 (4.1)	4 (2.1)	7 (3.6)	8 (4.1)	11 (5.6)	5 (2.6)
	Pork	143	6 (4.2)	6 (4.2)	6 (4.2)	6 (4.2)	6 (4.2)	-	Pork	173	9 (5.2)	6 (3.5)	4 (2.3)	6 (3.5)	14 (8.1)	6 (3.5)	4 (2.3)
	Total	237	8 (3.4)	7 (3.0)	7 (3.0)	6 (2.5)	6 (2.5)	1 (0.4)	Total	368	21 (5.7)	14 (3.8)	8 (2.2)	13 (3.5)	22 (6.0)	17 (4.6)	9 (2.4)
Cambodia	Pig	45	-	-	-	-	-	-	Pig	79	2 (2.5)	2 (2.5)	-	2 (2.5)	11 (13.9)	9 (11.4)	5 (6.3)
	Pork	76	-	-	-	4 (5.3)	4 (5.3)		Pork	57	1 (1.8)	1 (1.8)	-	-	7 (12.3)	4 (7.0)	5 (8.8)
	Total	121	-	-	-	4 (3.3)	4 (3.3)		Total	136	3 (2.2)	3 (2.2)	-	2 (1.5)	18 (13.2)	13 (9.6)	10 (7.4)
Lao PDR	Pig	30	-	-	-	-	-	-	Pig	79	6 (7.6)	2 (2.5)	4 (5.1)	5 (6.3)	18 (22.8)	16 (20.3)	6 (7.6)
	Pork	56	1 (1.8)	1 (1.8)	-	1 (1.8)	1 (1.8)	-	Pork	54	-	1 (1.9)	-	-	20 (37.0)	17 (31.5)	6 (11.1)
	Total	86	1 (1.2)	1 (1.2)	-	1 (1.2)	1 (1.2)	-	Total	133	6 (4.5)	3 (2.6)	4 (3.0)	5 (3.8)	38 (28.6)	33 (24.8)	12 (9.0)
Myanmar	Pig	1	-	-	-	-	-	-	Pig	71	13 (18.3)	10 (14.1)	3 (4.2)	5 (7.0)	-	2 (2.8)	-
	Pork	18	-	-	-	1 (5.6)	1 (5.6)	-	Pork	59	5 (8.5)	4 (6.8)	1 (1.7)	1 (1.7)	2 (3.4)	3 (5.1)	-
	Total	19	-	-	-	1 (5.6)	1 (5.3)	-	Total	130	18 (13.8)	14 (10.8)	4 (3.1)	6 (4.6)	2 (1.5)	5 (3.8)	-
	Grand total	463	9 (1.9)	8 (1.7)	7 (1.5)	13 (2.8)	12 (2.6)	1 (0.2)	Grand total	767	48 (6.3)	34 (4.4)	16 (2.1)	26 (3.4)	80 (10.4)	68 (8.9)	31 (4.0)

^1^ ESBL genes were examined in all ESBL producers. ^2^
*mcr* genes were tested in all isolates.

**Table 4 antibiotics-10-00657-t004:** Cephalosporin resistant *Salmonella* (*n* = 463) and *E. coli* (*n* = 767) from pig and pork.

Country	Source	*Salmonella*				*E. coli*			
		Total No.	No. of Positive (%)			Total No.	No. of Positive (%)		
			Ceftazidime	Cefotaxime	Cefpodoxime		Ceftazidime	Cefotaxime	Cefpodoxime
Thailand	Pigs	94	3 (3.2)	2 (2.1)	3 (3.2)	195	12 (6.2)	19 (9.7)	19 (9.7)
	Pork	143	5 (3.5)	6 (4.2)	6 (4.2)	173	14 (8.1)	15 (8.7)	16 (9.2)
	Total	237	8 (3.4)	8 (3.4)	9 (3.8)	368	26 (7.1)	34 (9.2)	35 (9.5)
Cambodia	Pigs	45	1 (2.2)	1 (2.2)	1 (2.2)	79	3 (3.8)	4 (5.1)	2 (2.5)
	Pork	76	-	-	-	57	4 (7.0)	1 (1.8)	1 (1.8)
	Total	121	1 (0.8)	1 (0.8)	1 (0.8)	136	7 (5.1)	5 (3.7)	3 (2.2)
Lao PDR	Pigs	30	1 (3.3)	1 (3.3)	1 (3.3)	79	3 (3.8)	8 (10.1)	7 (8.9)
	Pork	56	1 (1.8)	1 (1.8)	1 (1.8)	54	1 (1.9)	1 (1.9)	1 (1.9)
	Total	86	2 (2.3)	2 (2.3)	2 (2.3)	133	4 (3.0)	9 (6.8)	8 (6.0)
Myanmar	Pigs	1	-	-	-	71	7 (9.9)	13 (18.3)	13 (18.3)
	Pork	18	1 (5.6)	-	-	59	5 (8.5)	8 (13.6)	8 (13.6)
	Total	19	1 (5.6)	-	-	130	12 (9.2)	21 (16.2)	21 (16.2)
	Grand total	463	12 (2.6)	11 (2.4)	12 (2.6)	767	49 (6.4)	69 (9.0)	67 (8.7)

**Table 5 antibiotics-10-00657-t005:** Primers used in this study.

Gene	Primer	Primer Sequence (5′-3′)	Tm (°C)	Reference
MCR				
*mcr-1*	CLR-5F CLR-5R	CGGTCAGTCCGTTTGTTC CTTGGTCGGTCTGTA	58	[13]
*mcr-2*	MCR2-IF MCR2-IR	TGTTGCTTGTGCCGATTGGA AGATGGTATTGTTGGTTGCTG	58	[39]
*mcr-3*	MCR3-IF MCR3-IR	AAATAAAAATTGTTCCGCTTAT GAATGGAGATCCCCGTTTTT	58	[49]
ESBL				
*bla* _CTX-M_	CTX-M up CTX-M down	ATGTGCAGYACCAGTAARGTKATGGC TGGGTRAARTARGTSACCAGAAYCAGCGG	60	[50]
*bla* _TEM_	TEM up TEM down	GCGGAACCCCTATTTG TCTAAAGTATATATGAGTAAACTTGGTCTGAC	50	[51]
*bla* _SHV_	SHV up SHV down	TTCGCCTGTGTATTATCTCCCTG TTAGCGTTGCCAGTGYTCG	50	[51]
*bla* _CMY-1_	CMY1 up CMY1 down	GTGGTGGATGCCAGCATCC GGTCGAGCCGGTCTTGTTGAA	58	[51]
*bla* _CMY-2_	CMY2 up CMY2 down	GCACTTAGCCACCTATACGGCAG GCTTTTCAAGAATGCGCCAGG	58	[51]
*bla* _PSE_	PSE up PSE down	GCTCGTATAGGTGTTTCCGTTT CGATCCGCCGATGTTCCATCC	55	[50]
*bla* _CTX-M1_	MultiCTXMGp1-F MultiCTXMGp1-R	TTAGGAARTGTGCCGCTGYA CGATATCGTTGGTGGTRCCAT	688	[52]
*bla* _CTX-M2_	MultiCTXMGp2-F MultiCTXMGp2-R	CGTTAACGGCACGATGAC CGATATCGTTGGTGGTRCCAT	404	[52]
*bla* _CTX-M8/25_	MultiCTXMGp8/25-F MultiCTXMGp8/25-R	AACRCRCAGACGCTCTAC TCGAGCCGGAASGTGTYAT	326	[52]
*bla* _CTX-M9_	MultiCTXMGp9-F MultiCTXMGp9-R	GTGACAAAGAGAGTGCAACGG ATGATTCTCGCCGCTGAAGCC	850	[53]
*bla* _CTX-M15_	CTX-M-15-SF CTX-M-15-SR	CACACGTGGAATTTAGGGACT GCCGTCTAAGGCGATAAACA	996	[54]

## Data Availability

The data presented in this study are available in the article.

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
