# Peer review of "Colistin Resistance and ESBL Production in Salmonella and Escherichia coli from Pigs and Pork in the Thailand, Cambodia, Lao PDR, and Myanmar Border Area"

_antibiotics, 2021, doi:10.3390/antibiotics10060657_

Round 1

Reviewer 1 Report

The manuscript in its current form has a registration nature only, regarding the level of resistance to colistin and 3rd and higher generation cephalosporins in 2017-2018. It gives the impression that it is a repetition of the research presented in reference no. 26. Therefore, it is advisable that the Authors indicate differences in relation to the previous results (even to the level and scope of ESBL) to indicate why the current work is important.

The question also arises:

- Has any legal framework been made since the last to the present study to reduce the amount of positive isolation of Salmonella serotypes?

- It would also be advised to highlight the current legal situation regarding testing for Salmonella - are there any mandatory testing for animals or animal products? The statement "under less resitriction" is not very informative.

 It is also interesting that  S. Enteritidis serotype was not found among Salmonella strains. What is the prevalence of this serotype in the studied region? While the isolation of commensal strains of E. coli is understandable in a some sense, the isolation of Salmonella from every second sample (especially in products directly intended for consumption, seems to be a huge problem). I believe that the discussion on the above-mentioned topics would significantly change the registration character of the manuscript and would increase its scientific value.

Lines 58-65- A schematic map of the regions that the samples came from, would be very helpful here

Table 4: please correct the typo "resistant"

Section 4.6: The section concerning ESBL genes detection is not very clear

Line 280: Correct a typo (serotype name should be written in capital)

Reviewer 2 Report

The authors present a study examining prevalence of Salmonella and E.coli sourced from pigs and pork in select border countries. The methods and results are clearly described. Introduction and discussion sections would benefit from added detail. Itemized comments below.

  1. L 23: Authors should mention that Anatum was also a common serovar.
  2. L31: One Health is capitalized.
  3. L27: Add reference.
  4. L39: Add reference.
  5. L51: Authors should mention that for an organism to be classified as MDR, they must be resistant to three or more classes of antibiotics.
  6. L55: Be more specific about mcr variants. What is their significance? How do they vary?
  7. Table 1: Why is the subtotal for E.coli not given for each category? I know it is in the footnotes of the table but the authors should clarify as it is not well-understood by the reader.
  8. Table 1:Why is the subtotal for some categories in the Salmonella column not accurate? For instance, in location “Sa Kaeo”, there are 35 isolates in the subtotal row for Salmonella but 36 positives were identified.
  9. Tables 3 and 4: Authors should provide some discussion on how data in tables 3 and 4 are linked (i.e., does presence of ESBL-production always predict resistance to cephalosporins?)
  10. L127: This sentence is not clear.
  11. L135: Typo: “possive”
  12. L145-148: This is a very small results section and could be integrated into the above subsections.
  13. L151: Not sure if E.coli contamination is a great word choice. Would it still be considered “contamination” when E.coli is commensal to animals?
  14. L160-L162: What is the significance of this sentence? How does this impact the results?
  15. L163: Authors should not use the term “foodborne pathogens” as only one foodborne pathogen (Salmonella) was tested for in this study.
  16. L171-173: Be specific – what other antibiotics? Also, add reference.
  17. L176-178: What statistical test was used?
  18. L191: Authors should include some discussion of colistin usage in animal agriculture in these countries. In North America, colistin is not routinely used.
  19. L193-194: Not sure cross-contamination would be a big contributer to prevalence of colistin-resistant Salmonella in pork products, as this would suggest that large densities of colistin-resistant Salmonella is naturally present in the environment. I don’t believe this to be the case.
  20. L197: I believe the authors mean “former”, not the “latter”.
  21. L211: The One Health concept refers to the approach to achieving optimal human health through utilization of multiple sectors and disciplines. The way the authors have phrased this sentence is confusing and does not make sense.
  22. L294-300: The authors should include a short phrase about contamination at the retail level as well, since this was discussed in the discussion and supports some of the observations made.

Round 2

Reviewer 1 Report

I recommend the manuscript for publication in its current form.

Reviewer 2 Report

Comments have been addressed.